# Oral Health-Related Quality of Life (OHRQoL), Pain and Side Effects in Adults Undergoing Different Orthodontic Treatment Modalities: A Systematic Review and Meta-Analysis

**DOI:** 10.3390/healthcare13243317

**Published:** 2025-12-18

**Authors:** Ama Johal, Brian Dunne, Honieh Bolooki, Cara Sandler

**Affiliations:** Centre for OroBioengineering, Institute of Dentistry, Queen Mary University of London, London E1 4NS, UK

**Keywords:** oral health, quality, life, pain, side effects, adults, orthodontic, treatment, braces, aligners

## Abstract

**Background**: The present study aimed to identify the differences between experiences, in terms of oral health-related quality of life, pain, side effects and/or other complications, of adults undergoing orthodontic treatment using removable aligners and fixed labial or lingual appliances. **Methods**: The review was registered with PROSPERO, and a comprehensive electronic search was undertaken without language or date restrictions. Randomised and non-randomised trials and prospective cohort and cross-sectional studies along with case series were included. The Cochrane Collaboration’s Risk of Bias 2 Tool, Newcastle–Ottawa Scale and The Risk Of Bias In Non-Randomized Studies—of Interventions tools were used to assess quality. Data were grouped in terms of oral health-related quality of life, pain side effects and/or other complications. **Results**: Data from 35 studies were included; 9 were eligible for meta-analysis. Thus 2611 participants were included related to removable aligners (n = 513), fixed labial (n = 1816) and lingual (n = 218) appliances or a combination (n = 64) of appliances. The standardised mean differences in visual analogue scale pain reports between 24 h and 7 days were −10.02 (95% CI: −11.13, −8.91) for aligners and −6.40 (95%CI: −10.42, −2.38) for labial appliances (*p* = 0.09). There was a significant improvement in dental self-confidence following fixed labial appliance treatment (*p* = 0.001). **Conclusions**: No difference was detected in short-term pain with aligners and labial appliances. Aligners may have less impact on oral health-related quality of life measures compared to labial appliances. Lingual appliances have a persistent impact on speech, despite some adaptability. Any deterioration in oral health-related quality of life measures during treatment appears temporary. Further randomised trials using validated assessment tools and comparing aligners and labial and lingual appliances are required.

## 1. Introduction

With the increased demand for adult orthodontic treatment, orthodontics has evolved to facilitate the corresponding demand for appliances that are aesthetically pleasing and have little impact on everyday lives [1]. Generational differences may influence patients’ decisions to seek aesthetically pleasing appliances, with millennials reported to be more focused on image compared to previous generations [2].

Clinician-based outcomes have traditionally been the focus of much orthodontic research, with patient-based measures not being reported as frequently in the literature [3]. The drive of manufacturers to fabricate aesthetically pleasing appliances needs to be met with research evaluating these new appliances in terms of both clinician- and patient-based outcome measures.

Optimal oral health enables an individual to eat, speak and socialise without active disease, discomfort or embarrassment, which contributes to general well-being [4]. Quality of life (QoL) is defined as a person’s sense of well-being that stems from satisfaction or dissatisfaction with the areas of life that are important to them [5]. The impact of oral health and disease on QoL is known as oral health-related quality of life (OHRQoL). OHRQoL changes during treatment warrant more understanding to help the clinician understand how patients cope with the treatment [6]. Treatment experience is influenced by functional limitation, physical pain, psychological discomfort, physical disability, psychological disability, social disability and other disabilities [7].

An in-depth understanding of adult patient experiences with the different range of appliances now available would allow clinicians to provide more relevant information to patients and, in turn, facilitate the development of a patient-centred approach to providing more optimal care and informed consent. However, to date, there is relatively very little understanding of the impact of these appliance choices on their daily lives, with only two quantitative studies comparing the three appliance systems in relation to pain and oral health-related quality of life (OHRQoL) in adults, with their findings being inconclusive [8,9].

In contrast, a greater number of studies exist to compare clear aligner therapy with fixed labial appliances. They report participants treated with the former report less pain, eating difficulties and an overall lower in-treatment negative experience during the early stages of treatment [10,11,12,13]. In contrast, comparison of fixed labial and lingual appliances does not appear to demonstrate any significant differences in pain experience, but the latter were associated with greater oral discomfort, dietary changes, swallowing and speech disturbances and social problems than the labial appliance [14,15].

In light of the increasing uptake of adult orthodontic treatment and the availability of aesthetically pleasing alternative appliances, there is a need to systematically review the research in this field, in particular taking into account patient-centred outcomes. Thus, the present study aimed to identify the differences between experiences, in terms of oral health-related quality of life, pain, side effects and/or other complications, of adults undergoing orthodontic treatment using removable aligners and fixed labial or lingual appliances.

## 2. Methods and Materials

### 2.1. Protocol and Registration

This systematic review was registered with PROSPERO international prospective register of systematic reviews (www.crd.york.ac.uk/PROSPERO, accessed on 1 July 2020, CRD42020192841). The current systematic review and meta-analysis was carried out according to the Preferred Reporting Items for Systematic Reviews and Meta-Analyses (PRISMA) guidelines [8].

### 2.2. Eligibility Criteria

The following PICO selection criteria were applied:

Participants: Adults (older than 18 years) requiring orthodontic treatment (extraction or non-extraction).

Intervention: Orthodontic treatment with fixed (labial or lingual) appliances or removable (clear aligner) appliances.

Comparator: A comparison and/or control group was not essential for inclusion.

Outcomes:(i)Participant experiences and the impact of the various appliances on OHRQoL and self-esteem.(ii)Severity and nature of pain or discomfort experienced with orthodontic appliances.(iii)Other side effects and/or complications, including the effect on function.

The following were considered eligible: Randomised controlled trials (RCTs), non-randomised controlled clinical trials (CCTs), observational cohort and cross-sectional studies, prospective case series (minimum sample size of 10 patients) and qualitative studies exploring adults’ views and experiences were also included.

Exclusion criteria: Craniofacial syndromes, temporomandibular dysfunction, those prescribed analgesics or antidepressant medication for psychiatric disease or chronic medical conditions and participants undergoing dento-alveolar or orthognathic surgery.

### 2.3. Information Sources, Search Strategy and Study Selection

The search strategy is described in Appendix B Table A1. Comprehensive searches, without date or language restrictions, were conducted using the following electronic databases: MEDLINE via PubMed and Ovid, Web of Science, Cochrane and Embase. Databases were searched from inception until 3 May 2023. Unpublished or “grey” literature was searched using Google Scholar, OpenGrey, Directory of Open Access Journals, Digital Dissertations and the Meta-Register of Controlled Trials. Hand searching was performed from the reference lists of the full-text articles and other relevant systematic reviews. Full-text assessments of the studies for inclusion in the review were performed independently and in duplicate by two authors (HB and CS), and any disagreements were resolved by a third reviewer (AJ). If further information was required, the authors were contacted for clarification.

### 2.4. Risk of Bias and Quality Assessment in Individual Studies

The risk of bias assessment was performed independently and in duplicate by two authors (HB; CS), and any disagreements were resolved with a third reviewer (AJ). Different tools were used to assess research quality due to the diversity of study designs included. Randomised trials were assessed using the Cochrane Risk of Bias tool 2 (RoB2) [16].

The Newcastle–Ottawa Scale (NOS) was used to assess the quality of non-randomised studies including cohort studies [17]. An appropriately adapted version of the scale suitable for the assessment of cross-sectional studies was used for studies of this nature.

The Risk of Bias In Non-Randomised Studies—of Interventions (ROBINS-I) tool was used to assess one quasi-experimental study [18].

Studies with low or unclear risk of bias or medium to high quality were chosen for inclusion in any subsequent meta-analysis.

### 2.5. Data Items and Collection

The following characteristics were recorded: study design, sample size, participant details, treatment modality and outcome measures, e.g., OHRQoL, self-esteem, pain, side effects and other complications. Data was extracted and described according to the treatment modality (removable aligners, fixed labial or lingual appliances). Data was managed using Covidence© online systematic review software (https://www.covidence.org, accessed on 1 July 2020, Veritas Health Innovation Ltd, Melbourne, Australia).

### 2.6. Summary Measures and Approach to Statistical Analysis

OHRQoL scores reporting on the same domain at common timepoints were combined to obtain pooled mean proportion values, with standard deviation and/or 95% confidence intervals (CIs) if applicable. Visual analogue scale (VAS) scores for pain, when recorded at the same time interval, were managed in a similar manner. Data from qualitative studies was planned for synthesis if the same outcome was reported in more than two studies, followed by integration of quantitative and qualitative results.

### 2.7. Additional Analysis

A meta-analysis was carried out on studies with low/unclear risk of bias or with moderate/high quality, where the study designs were similar and data could be grouped. Cochrane’s Review Manager (version 5.4.1, available at revman.cochrane.org, accessed 07/2025) was used for meta-analysis, applying the inverse variance method with random effects and testing for the standardised mean difference. Results are presented as forest plots with weighted values and 95% CIs, with a *p*-value of less than 0.05 being considered statistically significant. The I^2^ statistic test was applied to quantify heterogeneity among studies.

## 3. Results

### 3.1. Study Selection and Characteristics of Included Studies

In total, 11,890 articles were identified, and following removal of duplicates, 11,157 studies were screened for eligibility. Title and abstract screening resulted in the identification of 118 articles which were suitable for full-text review. Subsequently 83 articles were excluded for not meeting the PICO selection criteria (Figure 1) and 35 studies were included, consisting of 6 randomised clinical trials, 1 controlled clinical trial, 1 quasi-experimental study, 23 prospective cohort studies, 2 cross-sectional studies, 1 longitudinal observational study and 1 qualitative study (Table 1). Nine studies met the inclusion criteria for meta-analysis, as illustrated in the PRISMA flow diagram (Figure 1).

A total of 2752 participants were identified from the 35 included studies. Two studies published a variety of outcomes from the same cohort [14,15] and a cross-sectional study by Flores-Mir et al. [10] included the data from the same Invisalign^®^ cohort as Pacheco-Pereira et al. [35]. Of the 2611 independent participants included, 1816 were treated with labial appliances, 513 with aligners (of which 307 were Invisalign^®^) and 218 with lingual appliances (of which 86 were Incognito). A further 52 participants were managed with a combination of labial appliances and Invisalign^®^ aligners, while 12 participants were included in a study involving thermoplastic retainers with appliances bonded to their lingual aspect for speech assessment.

### 3.2. Risk of Bias Within Studies

RoB2 was used to assess six studies (Figure 2a), NOS was used to assess 25 studies (Appendix B Table A2) with an adapted version used for two cross-sectional studies (Appendix B Table A3) and the ROBINS-I tool was used to assess one quasi-experimental study (Figure 2b). The one qualitative study was not assessed for risk of bias as this is not a commonly undertaken approach.

### 3.3. Results of Individual Studies, Meta-Analysis and Additional Analysis

The findings were reported in a descriptive manner as the wide variety in appliance type, outcome measures and timepoints meant that quantitative data could not be described in a meaningful way in the data extraction table (Table 1).

### 3.4. Qualitative Analysis of Results

#### 3.4.1. Three-Arm Studies (Aligners, Labial and Lingual Appliances)

Five studies compared the three treatment modalities in relation to pain and OHRQoL [9,27,28,29,41]). Findings were inconclusive across these studies as Shalish et al. [41] concluded pain was most severe in the lingual appliance group, in contrast to Antonio-Zancajo et al. (2020) [9] who reported lower levels of pain with lingual appliances. Antonio-Zancajo et al. [28] also reported that pain was mild in all groups apart from the labial appliance group, where pain was mild/moderate.

Zamora-Martinez [29] reported that QoL decreased significantly in all groups at the start of treatment but increased at the end of treatment. Alseraidi et al. [27] found that the aligner group reported the highest QoL scores and the labial appliance reported the lowest QoL scores.

#### 3.4.2. Two-Arm Studies

Aligners and labial appliances

Six publications described how participants with aligners experienced less pain than those with labial appliances [11,12,13,23,33,40]. Miller et al. [12] and Gao et al. [13] concluded that participants with aligners reported fewer negative impacts on overall OHRQoL during the initial phase of treatment.

Aligners appeared to be associated with lower in-treatment negative experiences, compared with fixed labial appliances during the first two weeks of treatment [10,11,12,13]. Four studies [10,11,19,20] described how those with fixed appliances had more difficulty eating and chewing compared to aligners.

Labial and lingual appliances

Two publications using the same participant cohort [14,15] found no significant difference in pain among those treated with labial or lingual appliances and that pain decreased for both groups over three months.

Participants with lingual appliances reported more oral discomfort, dietary changes, swallowing difficulty, speech disturbances and social problems than those with labial appliances [14,21].

#### 3.4.3. Single-Arm Studies

Labial appliances

Three studies reported pain during fixed appliance treatment [22,24,34]. Shen et al. [24] and Gibreal et al. [22] reported that pain peaks at 24 h while Johal et al. [34] described how pain peaked between 24 h and 3 days following appliance fit.

Three cohort studies used OHIP-14 to evaluate changes in OHRQoL [36,38,39]. Participants reported a negative effect on OHRQoL during treatment, but OHIP-14 scores returned to pre-treatment norms following appliance removal [36,39]. Two studies used OHIP-14 to assess OHRQoL [13,25] and showed a significant improvement at the end of treatment.

Several studies investigated the influence of labial appliances on OHRQoL using the Psychosocial Impact of Daily Aesthetics Questionnaire (PIDAQ) [31,32,37,45]. Psychological impact was shown to improve over the course of fixed appliance treatment with two authors reporting this as early as six months into treatment [31,37]. Studies that followed their participants for their entire course of orthodontic treatment showed improvements in all domains post-treatment [31,32].

Changes in self-esteem were measured using the Rosenberg self-esteem scale in three studies [36,39,45]. Johal et al. [39] reported no significant changes, while Choi et al. [36] reported significantly lower values 12 months into treatment. Both studies demonstrated improvements in participant scores following removal of appliances, with Choi et al. [36] reporting a level similar to pre-treatment scores and Johal et al. [39] concluding there were improvements in self-esteem. Varela & Garciacamba [44] reported no significant changes in self-esteem; however, these authors used the Tennessee Self Concept Scale which may not be comparable to the Rosenberg self-esteem scale.

Lingual appliances

Two studies investigated the influence that lingual appliances have on speech [42,43] and found that smaller profile lingual brackets had less negative influence. The investigators concluded there was a significant deterioration in speech, which improved at three months but was still worse than pre-treatment.

### 3.5. Aligners

Pacheco-Pereira et al. [35] reported high levels of satisfaction in eating and chewing which were attributed to the removable nature of aligners. Al Nazeh [30] found that aligner treatment had a less negative impact on OHRQoL in females.

### 3.6. Quantitative Analysis of Studies

A total of nine studies met the criteria for a meta-analysis [12,13,22,24,31,32,34,36,39].

### 3.7. Pain Experience

Figure 3 shows a forest plot with the combined VAS pain scores for studies that met the inclusion criteria. Aligners showed a significant reduction in pain scores between 24 h and 7 days after appliance fit, with an estimated reduction of −10.02 (95% CI: −11.13, −8.91). The two aligner studies were homogenous (I^2^ = 0%), adding significantly to the validity of the findings. The five labial appliance studies had high heterogeneity (I^2^ = 99%). The standardised mean difference in pain VAS scores between 24 h and 7 days was estimated −6.4 (95% CI: −10.42, −2.36) for labial appliances. The test for significant differences between them was not confirmed (χ^2^ = 2.89, *p* = 0.09). The heterogeneity for this comparison was moderate (I^2^ = 65.4%), suggesting a greater degree of caution in interpreting the findings from the studies evaluating labial appliances.

### 3.8. Oral Health-Related Quality of Life and Self-Esteem

Four studies reported pre- and post-treatment experiences [31,32,36,39]. There was no difference pre-treatment when comparing aligners and labial appliances (Std. mean difference = 0.0, 95% CI: −1.5, 1.5; Figure 4). However, at all other timepoints a significant difference was found, with aligners scoring significantly lower than labial appliances (Figure 4). Lingual and straight wire appliances were also compared (Figure 5). There was no difference between appliances pre-treatment (Std. mean difference = 0.4, 95% CI: −3.0, 3.7). Statistically significant differences were observed at 1-week (Std. mean difference = 7.2, 95% CI: 4.3, 10.0) and 1-month (Std. mean difference = 3.8, 95% CI: 1.4, 6.1). Where a difference was observed, OHRQoL scores were significantly higher for the lingual appliance. Where heterogeneity could be assessed, there was relatively little heterogeneity in terms of sizes of group differences.

Choi et al. [36] indicated a non-significant reduction over time in Rosenberg self-esteem scale measurements (Std. mean difference = −0.16, 95% CI: −0.5, 0.18), while Johal et al. [39] indicated a significant increase (Std. mean difference = 0.45, 95% CI: −0.04, 0.86) (Figure 6). The pooled data test was insignificantly positive.

Two studies reported changes in OHRQoL using the PIDA [31,32]. The standardised mean difference for dental self-confidence tested across two publications indicated a significant increase between pre- and post-treatment scores of 2.18 (95% CI:0.87, 3.5) (z = 3.25, *p* = 0.001) (Figure 7). However, the heterogeneity was high between the studies (I^2^ = 97%), suggesting a greater degree of caution in interpreting the findings from these studies.

A reduction in aesthetic concern was not significant when the data were pooled (Figure 8), with scores of −1.71 (95% CI: −3.46, 0.04) (z = 1.91, *p* = 0.06). Heterogeneity across the studies was very high (I^2^ = 99%), again suggesting a greater degree of caution in interpreting the findings from these studies.

## 4. Discussion

Pain experienced by adults undergoing orthodontic treatment with aligners and labial appliances was not statistically significant in this meta-analysis (Figure 3; *p* = 0.09). Many low quality or high risk of bias studies reported that aligners were less painful than labial appliances; these were not included in the quantitative analysis [11,23,33,40].

This systematic review did not conclude whether labial or lingual appliances inflict more pain during orthodontic treatment. The differences reported between studies may be related to the lack of randomisation and potential confounding factors. Meta-analysis of these studies was deemed inappropriate due to shortcomings in study design. A recent qualitative study looking at the motivations for treatment, choice and impact of aligners and labial and lingual appliances also reported similar experiences in adults [47].

Selection criteria varied widely between studies. The need for dental extractions may influence pain experienced by a participant undergoing orthodontics and also the overall treatment experience.

It is evident from this review that all adult orthodontic treatment will inflict a certain degree of pain and orthodontic pain was commonly described to peak at 24 h following appliance fit. Irrespective of appliance, pain was reported to be less severe following subsequent visits for adjustments [15,23,34,40]. This change likely represents participants becoming accustomed to the experience of orthodontic treatment as it progressed.

Aligners appear to outperform labial appliances within the first two weeks of treatment in terms of in-treatment participant experiences [8,11,12,13]. Data available is short-term and there is little longitudinal data comparing OHRQoL between participants treated with aligners and other orthodontic treatment modalities.

Aligners were used to manage relatively mild malocclusions on a non-extraction basis, while fixed appliances are routinely used in conjunction with extractions, if deemed necessary. One would expect participants treated on an extraction basis to report more negative influences on function secondary to food packing; however, the studies included in this review did not develop participant experiences with this in mind. These are important confounding variables that may adversely affect the conclusions of the individual investigations.

This review suggests that lingual appliances are problematic in terms of speech and tongue soreness [8,42,43]. Longitudinal data reported by Hohoff et al. [42] and Wu et al. [14] describes how speech improves throughout the course of treatment but is still substantial several months into treatment. These findings are supported by a recent qualitative study [47] and patients should be counselled regarding this potential treatment experience.

Several studies reported the negative impact labial appliances have on OHRQoL during treatment, followed by an improvement after removal of the appliance. Meta-analysis of the pre- and post-treatment results comparing labial appliances with aligners showed that at all timepoints OHRQoL scores for aligners were significantly lower than for labial appliances. In contrast, OHRQoL scores for the lingual appliance suggested a worse quality of life. Patients should be counselled on the potential negative sequalae that take place after appliance fit and these negative discussions may be balanced by informing the patient of positive changes in dental self-confidence that are likely to occur as result of treatment, as portrayed by the forest plot in Figure 7.

Three authors reported no significant deterioration in self-esteem during treatment with labial appliances [39,44,45] while one [36] described a worsening amongst their cohort. Following removal of the appliances, two studies reported no changes compared to pre-treatment self-esteem [36,44], while one study suggested a significant improvement [39]. Romero-Maroto et al. [45] did not follow their cohort until treatment was completed. The results of two studies that used the Rosenberg self-esteem scale were combined in a meta-analysis; however, only pre- and post- treatment scores were available as common timepoints [36,39]. Meta-analysis concluded that orthodontic treatment with labial appliances does not result in a statistically significant change in self-esteem, compared to pre-treatment (*p* = 0.66; Figure 6).

### Limitations

The findings may not be extrapolated to all clinical settings in which these appliances are used, for example, adolescent patients. The present study only evaluated the evidence in relation to adults seeking orthodontic treatment to identify any differences between their experiences with these appliances. Whilst systematic reviews offer high-level evidence, the quality of some included studies varied. However, retrospective studies were excluded due to high selection bias risk, but few studies used proper randomisation; issues like poor allocation concealment were common. Randomisation is challenging in adult orthodontics due to strong aesthetic preferences, risking dropout or refusal to participate if adults felt less aesthetically pleasing treatment options were being used. Within the review, studies showed high variability in populations, outcome measures, appliance types and diverse tools (e.g., VAS, OHIP-14, PIDAQ, Rosenberg scale), which limited comparability. The contents of the publication have been verified against the PRISMA checklist (see Appendix A).

## 5. Conclusions

There is a lack of consistency measuring in-treatment experiences with a number of both validated and non-validated questionnaires used to assess changes in OHRQoL in adults undergoing orthodontic treatment.From the evidence currently available, there appears to be no significant difference in the pain experienced by adults undergoing orthodontic treatment with aligners or labial appliances during the first week of treatment.Aligners may have less impact on OHRQoL when compared to labial appliances within the first two weeks of treatment, as their removable nature allows participants to eat and chew effectively.Despite some adaptability, lingual appliances have a persistent impact on speech throughout the duration of treatment.Any deterioration in OHRQoL measures with labial appliances during treatment is temporary, and these return to pre-treatment norms following appliance removal, with a statistically significant improvement in dental self-confidence.

## Figures and Tables

**Figure 1 healthcare-13-03317-f001:**
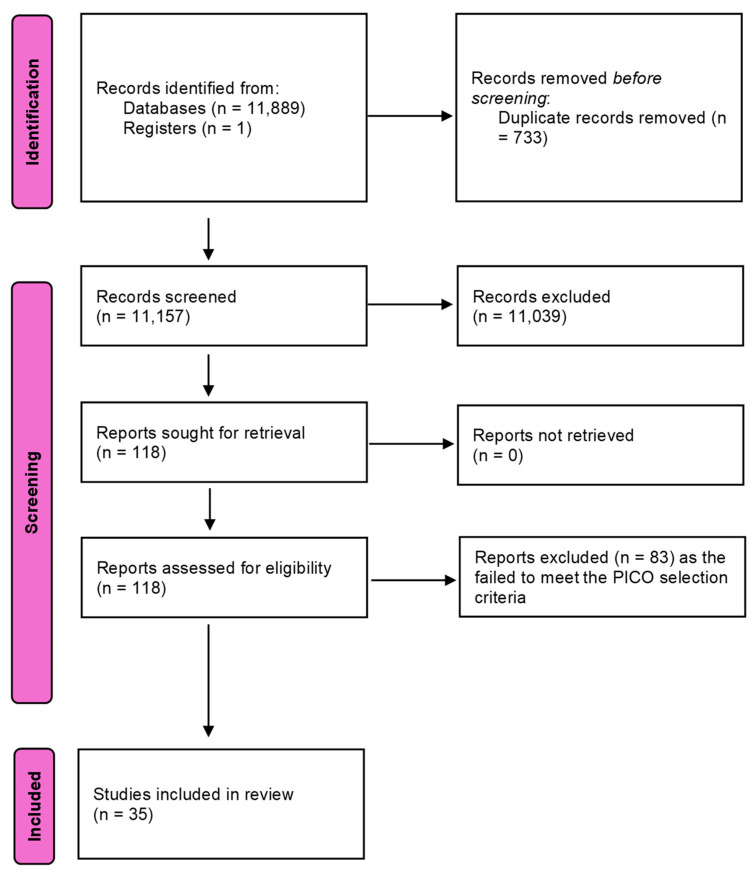
Preferred reporting items for systematic reviews and meta-analyses (PRISMA) flow diagram.

**Figure 2 healthcare-13-03317-f002:**
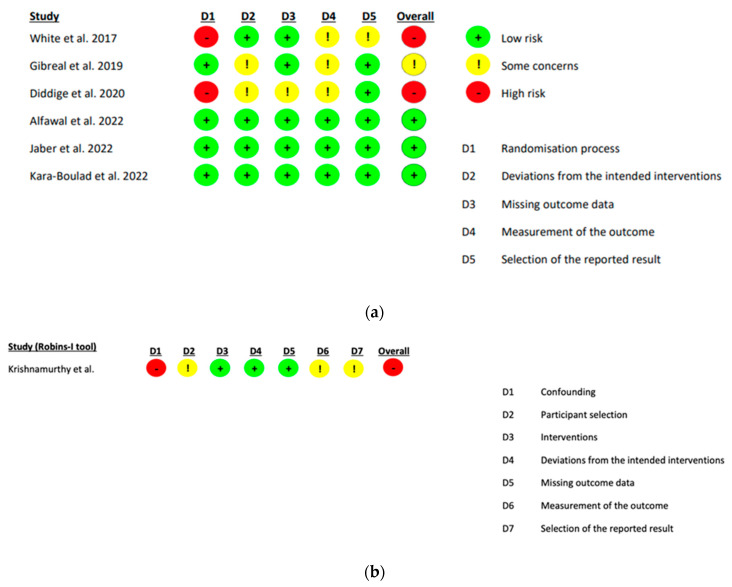
(**a**) Cochrane risk of bias 2 assessment of bias within randomised studies [11,19,20,21,22,23]. (**b**) ROBINS-I assessment tool for quasi-experimental studies [25].

**Figure 3 healthcare-13-03317-f003:**
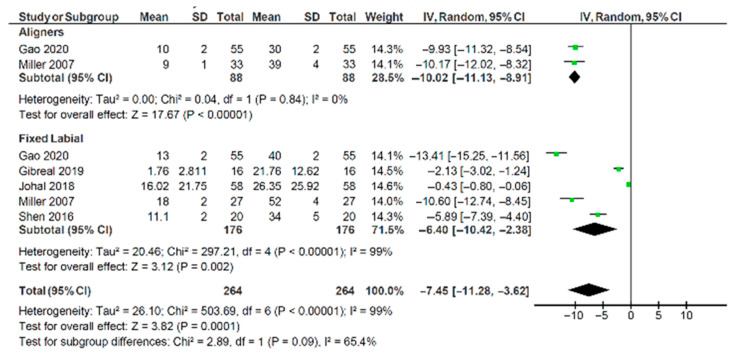
Forest plot illustrating pain visual analogue scale scores for aligners and labial appliances [12,13,22,24,34].

**Figure 4 healthcare-13-03317-f004:**
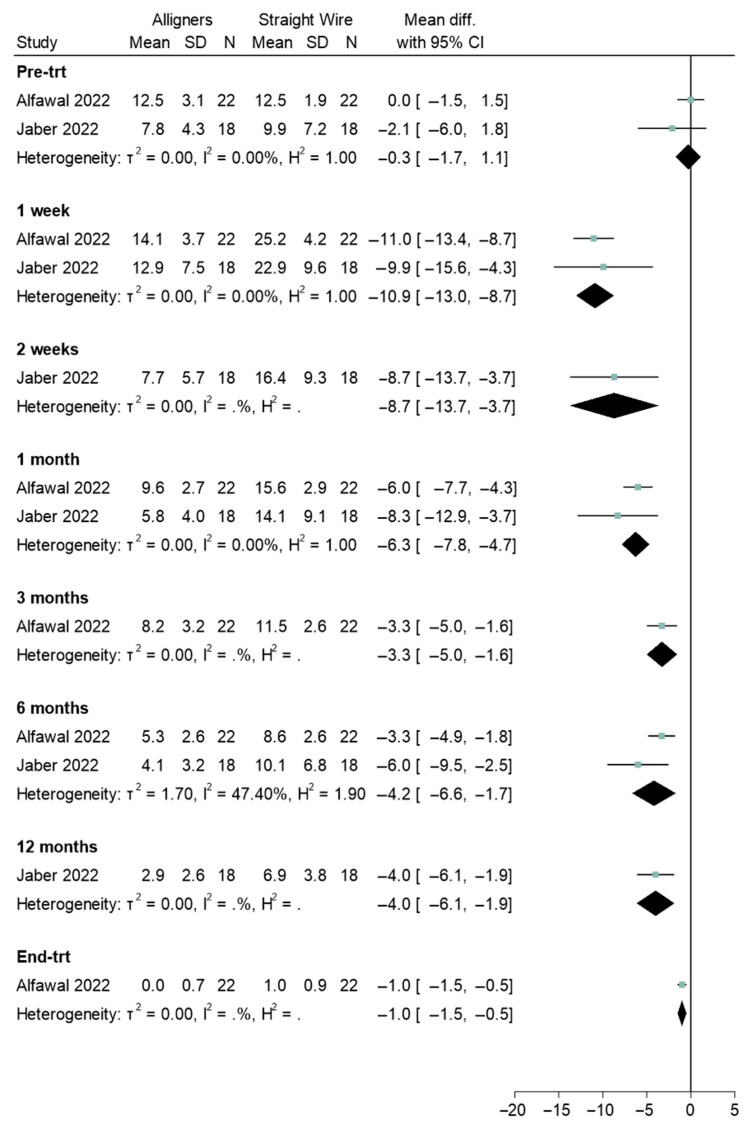
Forest plot illustrating the comparison of oral health-related quality of life (OHRQOL) scores between aligners and straight wire (labial) appliances [19,20].

**Figure 5 healthcare-13-03317-f005:**
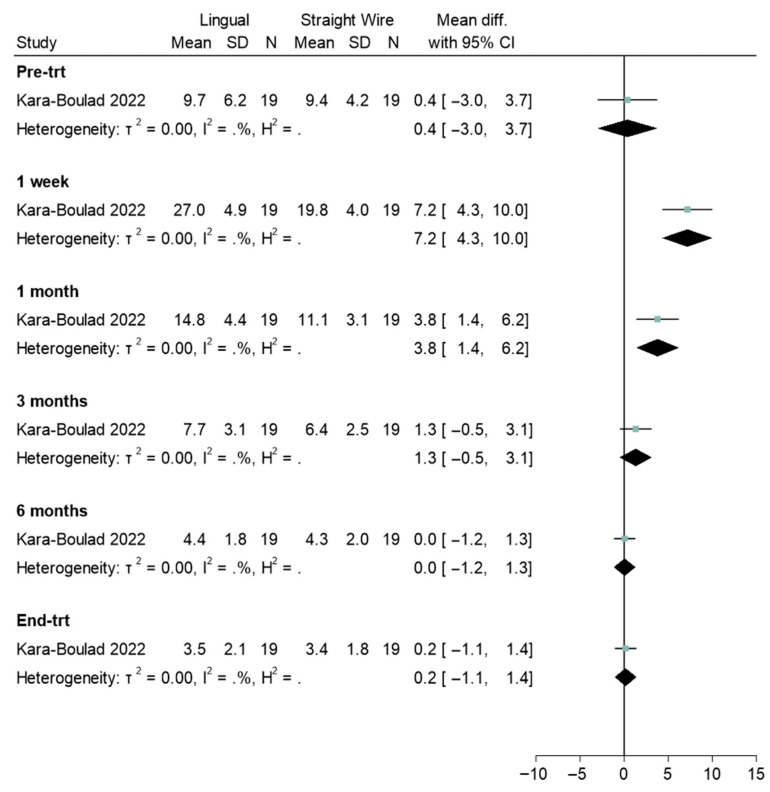
Forest plot illustrating the comparison of oral health-related quality of life (OHRQOL) scores between lingual and straight wire (labial) appliances [21].

**Figure 6 healthcare-13-03317-f006:**
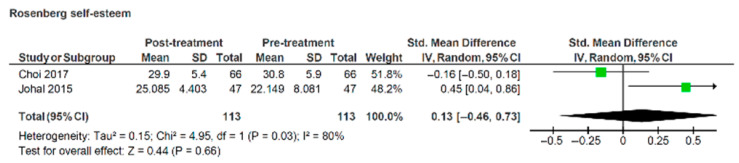
Forest plot illustrating Rosenberg self-esteem scores pre- and post-treatment with labial appliances [36,39].

**Figure 7 healthcare-13-03317-f007:**
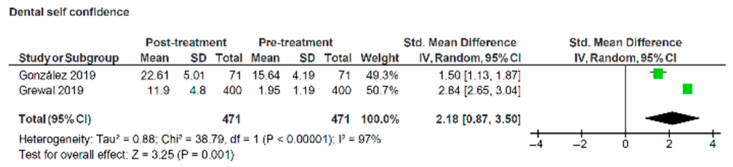
Forest plot illustrating changes in dental self-confidence domain of psychosocial impact of dental aesthetics questionnaire pre- and post- treatment with labial appliances [31,32].

**Figure 8 healthcare-13-03317-f008:**
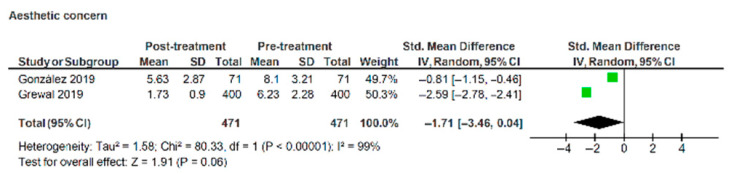
Forest plot illustrating changes in aesthetic concern domain of psychosocial impact of dental aesthetics questionnaire pre- and post- treatment with labial appliances [31,32].

**Table 1 healthcare-13-03317-t001:** Characteristics of included studies (n = 35).

Study ID		Sample Size Appliance(s) Setting	Case Type	Outcome Measure(s) Timepoints for Measurements	Findings
1	Alfawal et al. [19] RCT	n = 44 SWA (22) Aligners (22) Dental Hospital	Class I molar and canine relationships Mild-moderate crowding both arches Non-extraction cases No missing teeth	OHIP-14 Pre-treatment 7 days 1 month 3 months 6 months Post-treatment	OHRQoL of patients treated with aligners was significantly higher than those treated with fixed appliances during the whole period of treatment. Orthodontic treatment has a temporary adverse effect on OHRQoL in both groups, peaking one week following the start of treatment, then a gradual improvement was reported. OHRQoL was greatly improved at the end of orthodontic treatment regardless of the treatment modality. Aligners reduced the length of treatment by 26% compared to fixed appliances.
2	Jaber et al. [20] RCT	n = 36 SWA (18) Aligners (18) Dental Hospital	Class I malocclusion >5 mm crowding ≥25 ABO-DI No missing teeth No history of trauma	OHIP-14 Baseline 7 days 14 days 1 month 6 months 1 year	Both groups reported a functional limitation one week after appliance placement (*p* = 0.001). OHRQoL improves throughout treatment after initial phase of deterioration. Patients treated with clear aligners had less impact on OHRQoL than those treated with fixed appliances. Functional limitation, pain and physical disability were most affected aspects for both groups.
3	Kara-Boulad et al. [21] RCT	n = 38 SWA (19) Lingual (19) (DTC Orthodontics, Hangzhou, China) Dental Hospital	Class I molar and canine relationships 4–6 mm of crowding Non-extraction treatment No anterior crossbites No craniofacial syndromes	OHIP-14 Pre-treatment 1 week 1 month 3 months 6 months End of active treatment	OHRQoL improved in both groups after treatment. OHRQoL was better in the labial group during the first month of treatment. For both groups, the greatest deterioration of OHRQoL was in the first week of treatment but this decreased over time. Functional limitation, physical pain, physical disability and social disability were more common in the lingual group compared to the labial group. Psychological disability was significantly greater in the labial group (*p* < 0.05).
4	Diddige et al. [11] RCT	n = 36 SWA (12) SLBs (12) Aligners (Smile Align^®^, India) (12) Does not specify setting	Non-extraction Full complement teeth LII 3–5 mm	OHRQoL (Miller 2007 [12] questionnaire) Pain (VAS) Speech 4 h 24 h 3 days 7 days	Aligners cause less pain than fixed labial appliances during 1st week of treatment. (*p* = 0.001) Aligners result in fewer eating disturbances compared to fixed appliances (conventional and SLBs) (*p* < 0.001). Participants in the aligner group had maximum satisfaction in terms of appliance aesthetics. (*p* < 0.001) Analgesic consumption higher in conventional group compared to aligner or SLBs (*p* < 0.001).
5	Gibreal et al. [22] RCT	n = 16 SWA Dental Hospital	II division I malocclusions requiring premolar extractions Severe lower arch crowding	Pain (VAS) 1 day 7 days 14 days 28 days	No significant difference in pain levels described between control group (group included in this systematic review) and experimental group (group underwent piezocision-assisted orthodontics).
6	White et al. [23] RCT	n = 40 SWA (18) Aligners (22) (Invisalign^®^) Dental Hospital	Class I malocclusion Non-extraction Less than 4 mm crowding	Pain (VAS) Immediately after appliance fit and every day for the following seven days 4 days’ worth of measurements after first and second adjustments	Fixed labial appliances produce more discomfort than aligners (*p* = 0.039). Fixed labial appliance participants have more difficulty chewing (*p* = 0.023). Participants treated with both fixed labial and aligners report less pain at subsequent visits. Analgesia consumption mirrors levels of discomfort. (*p* < 0.05) on day two—more analgesia fixed labial group.
7	Shen et al. [24] CCT	n = 20 SWA (SLBs) Dental Hospital	Non-extraction	Pain (VAS) 2 h 24 h 7, 14 and 30 days	All participants reported pain induced by orthodontic force, which peaked at 24 h and dissipated over 7 days. No healthy controls reported pain at any timepoint.
8	Krishnamurthy et al. [25] Quasi-experimental	n = 34 SWA Dental Hospital	I/II/III malocclusions No missing teeth ANB −1° to 6°	OHIP-14 Pre-treatment 1-month post-debond	There was a significant reduction in OHIP-14 scores before and after treatment with fixed orthodontic appliances (*p* < 0.001). There was a significant improvement in OHRQoL following completion of fixed orthodontic treatment.
9	Lau et al. [26] Cohort	n = 92 SWA Dental Hospital	Adults undergoing fixed orthodontic treatment Dental disease, orthognathic cases, previous orthodontic treatment all excluded	OHIP-14 PAR ICON Pre-treatment Post-treatment	Orthodontic treatment had a positive and clinically important impact on the OHRQoL of patients. OHRQoL was significantly higher among those patients whose dental occlusion was categorised as ‘improved/greatly improved’ compared with those categorised as ‘worse/not different’, according to the PAR score (*p* = 0.03).
10	AlSeraidi et al. [27] Cohort	n = 117 SWA (41) Lingual (37) (Incognito^®^) Aligners (39) (Flash Orthodontics Mumbai, India) Specialist practice	Adults undergoing orthodontic treatment Crowding >8 mm, extractions, previous orthodontic treatment, oral pathology, significant medical history/medications, use of auxiliaries all excluded	WHOQOL-BREF Single questionnaire given between 6 and 9 weeks	Those treated with aligners had significantly better QoL scores, followed by the lingual group and the vestibular group. The aligner group obtained significantly higher scores for psychological health. Aligner and lingual scores for social relationships were significantly greater than those of the vestibular group. Environment displayed significant differences between all groups.
11	Antonio-Zancajo et al. [28] Cohort	n = 120 SWA (30) Low friction SWA (30) (Synergy^®^, USA) Lingual (30) (STB^®^, Ormco^®^) Aligners (30) (Invisalign^®^) Does not specify setting	18–40 years old I/II/mild III skeletal TSALD −6 to −2 mm No extractions (excluding third molars) Good oral and general health No prior orthodontic treatment No severe malformations No surgical treatment No medications/medical conditions influencing pain perception	Ortho-SF-MPQ 4 h 8 h 24 h 2 days 3 days 4 days 5 days 6 days 7 days	Most frequently affected pain location was both anterior arches, followed by the “anterior maxilla” and mandible in all groups. Pain was mild/moderate in the conventional labial appliance group and mild in the low friction, lingual and aligner groups in the first 24 h, progressively decreasing with greater speed in the lingual group. The most frequently reported pain in the first 24 h was acute pain in the low friction and lingual groups, and sensitivity in the conventional labial appliance and aligner groups. The location and amount of pain was similar in all groups.
12	Zamora-Martínez et al. [29] Cohort	n = 120 SWA (30) (metal brackets) SWA (30) (aesthetic/ceramic brackets) Lingual (30) Aligners (30) University dental clinic	Adults with good oral and general health Orthognathic surgery, previous orthodontic treatment, those who missed >3 appointments/did not complete protocol excluded	OHIP-14 Pre-treatment 6 months End of treatment	All groups had a significant reduction in OHRQoL during treatment compared to pre-treatment, and significant improvements in quality of life at end of treatment. The negative impact was greater in the group with labial appliances in the first 6 months. Almost all domains improved in all groups pre- and post-treatment, except functional limitation was unchanged in those with metal brackets and physical pain was unchanged in those with aesthetic/ceramic brackets and lingual appliances.
13	Al Nazeh et al. [30] Cohort	n = 50 Aligners (Invisalign^®^) Dental Hospital	Adults with no previous orthodontic, surgical, prosthodontic or implant treatment No pathology or treatment failure/clinical problems during investigation No medication/medical issues	OHIP NEO-FFI Pre-treatment Post-treatment	Aligner treatment has less negative oral health impacts in females but not males. Personality profiles contribute to the impact of treatment on OHRQoL differently between males and females. Openness before treatment and extraversion, openness and conscientiousness after treatment can predict the oral health impact of aligner treatment in males.
14	Antonio-Zancajo et al. [9] Cohort	n = 120 SWA (30) SWA SLBs (30) Aligners (30) Lingual (30) (STB^®^, Ormco^®^) Does not specify setting	18–40 years No previous orthodontics Non extraction 2–6 mm of crowding in both arches Skeletal I or mild II/III	Pain OHIP-14 Pain: 4, 8, 24 h and daily for 7 days OHIP-14: 1 month into treatment	Pain peaked between 24 and 48 h post appliance fit. Lingual appliance participants reported lower levels of pain at all times analysed, and their scores in the total OHIP-14 indicated less impact on their oral quality of life (*p* < 0.01).
15	Gao et al. [13] Cohort	n = 110 SWA (55) Aligners (55) Dental Hospital	Older than 18 years and receiving treatment in both arches All malocclusion types	Pain VAS STAI-S OHIP-14 Pain: Days 1–14 OHIP-14: Days 1, 7, 14 STAI: Days 1, 3, 5, 7, 14	Participants treated with clear aligners experienced lower pain levels (*p* < 0.05 on days 1, 2, 4 and 5), less anxiety (*p* < 0.05 at all timepoints [1, 3, 5, 7, 14 days]) and higher OHRQoL (*p* < 0.05 at all timepoints [1, 7, 14 days]) as compared those treated with fixed labial appliances.
16	González et al. [31] Cohort	n = 71 SWA Dental Hospital	Class I molars <3 mm crowding	PIDAQ Pre-treatment 6 months Debond	Dental self-confidence was shown to increase at both six months (*p* = 0.001) into treatment and post-treatment (*p* = 0.001). Psychological impact was shown to improve at both six months (*p* = 0.049) into treatment and post-treatment (*p* = 0.01). Aesthetic concern was shown to decrease between pre- and post- treatment (*p* = 0.031).
17	Grewal et al. [32] Cohort	n = 400 SWA “Two orthodontic centres”	IOTN “Definite need for treatment”	PIDAQ Pre-treatment Post-treatment	Statistically significant psychosocial impacts after orthodontic treatment were observed in all domains (dental self-confidence, social impact, psychological impact, aesthetic concern, functional limitation, matrimonial concerns).
18	Almasoud, [33] Cohort	n = 64 SWA (32) (Damon Q^®^) Aligners (32) (Invisalign^®^) Specialist practice	Class I molar relationship LII 3–5 mm	Pain VAS 4 h 24 h 3 days 7 days	Participants treated with aligners reported less pain during the first week of treatment compared to those treated with fixed labial appliances (SLBs) (*p* = 0.001).
19	Johal et al. [34] Cohort	n = 58 SWA 5 specialist practices	Extraction and non-extraction (6/51)	Pain (VAS) 4 h 24 h 3 days 7 days After appliance fit and following first two adjustment appointments	Pain peaked between 24 h and three days following appliance fit. Pain is less intense following participants second and third fixed appliance adjustment appointments (*p* < 0.001). Analgesic use mirrored pain experience. Dental irregularity, gender or age did not predict pain experience.
20	Pacheco-Pereira et al. [35] Cohort	n = 81 Aligners (Invisalign^®^) 4 specialist practices	“Adult participants treated exclusively with Invisalign”	Dental Impact of Daily Living Patient Satisfaction Questionnaire Post-treatment	Most significant improvements with treatment were in the appearance and eating and chewing categories. Positive doctor–participant relationship correlated with high levels of participant satisfaction. Food packing between teeth and pain were the most common sources of dissatisfaction. The negative experiences were not strong enough to reduce participants’ overall positive experience.
21	Choi et al. [36] Cohort	n = 66 SWA Dental Hospital	Extraction and non-extraction (15/51)	OHIP-14 STAI Zung self-rating depression scale Rosenberg self-esteem scale Key subjective food intake ability Baseline 12 months Debond	OHRQoL temporarily deteriorates during labial fixed appliance treatment (*p* = 0.002). This is due to an increase in psychological (*p* = 0.008) and social disabilities (*p* < 0.001). Changes in OHRQoL were associated with age, psychological health (anxiety, depression, self-esteem) and subjective food intake ability. OHRQoL recovered following treatment (*p* = 0.22).
22	Prado et al. [37] Cohort	n = 60 SWA Dental Hospital	Participants 18–30 requiring fixed appliances	PIDAQ Pre-treatment 6 months	The first six months of orthodontic treatment improve the psychosocial impact of malocclusion. Participants report a greater aesthetic impact [worse scores] (*p* < 0.001) and less psychological impact [better scores] after six months of fixed labial appliances (*p* < 0.001).
23	Chen et al. [38] Cohort	n = 190 SWA Dental Hospital	Any malocclusion excluding the need for orthognathic surgery or previous history of orthodontics	OHIP-14 Pre-treatment Post-treatment	Malocclusion has a significant negative impact on OHRQoL. Orthodontic treatment improved OHRQoL among adults (*p* < 0.05).
24	Johal et al. [39] Cohort	n = 61 SWA 4 Specialist practices	Participants of at least 18 years requiring fixed appliances Surgical cases excluded	OHIP-14 Rosenberg self-esteem scale Pre-treatment 1, 3 and 6 months Post-treatment	Fixed orthodontic appliance therapy causes a negative impact on OHRQoL during the first three months of treatment (*p* = 0.001). There is no significant difference between pre- and post-treatment OHRQoL scores (*p* = 0.078). OHRQoL scores return to pre-treatment norms after fixed appliance treatment. There is a significant increase in self-esteem as a result of fixed appliance treatment (*p* = 0.002).
25	Fujiyama et al. [40] Cohort	n = 145 SWA (55) Aligners (Invisalign^®^) (38) SWA and Aligners (52) Specialist practice	Extraction and non-extraction (30/115)	Pain VAS 60 s 6 h 12 h Each day for next 7 days Same as above at 3 weeks Same as above at 5 weeks	A significant difference was observed in overall VAS pain between fixed labial and Invisalign^®^ participants in intensity of pain (*p* < 0.05), number of days the pain lasted (*p* < 0.05), and discomfort level (*p* < 0.05), favouring Invisalign^®^.
26	Shalish et al. [41] Cohort	n = 68 SWA (28) Lingual (19) (Incognito^®^) Aligners (21) (Invisalign^®^) Dental Hospital and two private practices	“Consecutive adult patients who needed comprehensive orthodontic treatment”	Daily OHRQoL diary Pain (VAS) Recorded on each day from days 1–7 and again on day 14.	Lingual appliances were associated with more severe pain and analgesic consumption, the greatest oral and general dysfunction and the most difficult and longest recovery. Aligner participants complained of high levels of pain initially; however, this group had the lowest level of oral symptoms. Participants treated with aligners and labial appliances had similar levels of general activity disturbances and oral dysfunction. Many lingual and some buccal appliance participants did not reach a full recovery from eating difficulties by the end of the study.
27	Wu et al. [14] Cohort	n = 60 SWA (30) Lingual (30) (Incognito^®^, 3M Unitek, Bad Essen, Germany) Dental Hospital	Labial: 20.33 +/− 4.205 years Lingual: 21.63 +/− 2.236 years	Patient impacts (discomfort, mastication, speech, social function)—non validated Pain VAS 1 week 1 month 3 months	All participants experienced oral impact disturbances, which decreased over time (*p* < 0.001). Participants treated with customised lingual appliances reported more oral discomfort (*p* < 0.001), dietary changes (*p* < 0.001), swallowing difficulty (*p* < 0.001), speech disturbances (*p* < 0.001) and social problems (*p* < 0.001) than did those in labial appliance group. There was no significant difference between the groups regarding ratings of oral self-care, mastication and satisfaction level of treatment (*p* > 0.05).
28	Wu et al. [15] Cohort	n = 60 SWA (30) Lingual (30) (Incognito^®^, 3M Unitek, Bad Essen, Germany) Dental Hospital	Labial: 20.33 +/− 4.205 years Lingual: 21.63 +/− 2.236 years	Pain VAS 1 week after placement 1 month after placement 3 months after placement	No significant difference in global ratings of pain among those treated with labial or lingual appliances (*p* > 0.05). Global ratings of pain decreased for both groups over the study period (*p* < 0.001). Lingual appliances cause more tongue soreness (*p* < 0.001). Labial appliances cause more lip and cheek soreness (*p* < 0.001).
29	Miller et al. [12] Cohort	n = 60 SWA (27) Aligners (Invisalign^®^) (33) Private practice and Dental Hospital	Extraction and non-extraction Included re-treatment cases	Daily diary (amended version of the geriatric oral health assessment index) Pain VAS 4 h 24 h 3 days 7 days	Participants treated with Invisalign^®^ reported fewer negative impacts on overall OHRQoL (*p* < 0.001). Invisalign^®^ group reported less impact in each quality of life subscale (functional, psychosocial, pain-related) (*p* < 0.03). VAS pain scores show that Invisalign^®^ caused less pain during the first week of treatment compared to fixed labial appliances (*p* < 0.0001). Participants treated with Invisalign^®^ took less pain medication compared to fixed labial participants (*p* < 0.07).
30	Hohoff et al. [42] Cohort	n = 12 Thermoplastic aligner with lingual appliance Simulated environment	German speakers	Speech evaluation by speech professionals Subjective evaluation by participants Before placement 10 min 24 h	All lingual appliances led to significant impairments in sound performance and oral comfort, but with inter-appliance differences in the degree of impairment. Smaller lingual appliances result in less impairments.
31	Hohoff et al. [43] Cohort	n = 23 Lingual (7th Generation, Ormco^®^) Does not specify (Not relevant to outcomes)	French speakers Maxillary arch treatment only	Digital Sonography Speech Professionals evaluation Subjective evaluation Pre-treatment 24 h 3 months (+/−1 week)	The researchers concluded there was a significant deterioration in speech, which improved by three months somewhat but was still significantly worse than pre-treatment.
32	Varela & Garciacamba [44] Cohort	n = 47 SWA Dental Hospital	Moderate-severe malocclusion with aesthetic repercussions	Tennessee Self Concept Scale Secord and Jourards Cathexis Scale Pre-treatment 6 months Post-treatment	No significant changes in self-concept and self-esteem were observed after treatment. Increases for both subscales of body image, overall and facial, were observed. Overall body image demonstrated and improvement 6 months after the start of treatment.
33	Flores-Mir et al. [10] Cross-sectional	n = 122 SWA (41) Aligners (81) (Invisalign^®^) SWA: Dental Hospital Aligners: 4 specialist practices	Any malocclusion complexity	Dental Impacts on Daily Living Patient Satisfaction Questionnaire Debond	Participants treated with fixed labial appliances had statistically similar satisfaction outcomes across all domains analysed, except for eating and chewing when compared to Invisalign^®^. Invisalign^®^ participants reported more satisfaction in eating and chewing during treatment. Participant satisfaction remained similar six months after treatment irrespective of treatment modality.
34	Romero-Maroto et al. [45] Cross-sectional	n = 85 SWA Dental Hospital	I/II/III malocclusions Non extraction Crowding less than 6 mm	PIDAQ STAI-S Rosenberg self-esteem scale 3–6 months	The labial fixed appliance group showed significantly higher scores for social impact, psychological impact and aesthetic concern compared to the control group. No significant difference in self-esteem between the treatment and control groups. Anxiety plays a fundamental role in the effect of perceived dental impact on self-esteem in adult participants. Self-esteem correlated negatively with all dimensions of dental appearance impact [dental self-confidence, social impact, psychological impact, aesthetic concern], except for the positive dental self-confidence dimension that correlates positively.
35	Wong et al. [46] Qualitative	n = 26 SWA Specialist practice and Dental Hospital	Any fixed appliance treatment excluding orthognathic	Structured interview Post-treatment	One of the main five themes identified was impact of appliance treatment. Four subthemes identified under this heading including discomfort, function/oral hygiene, aesthetics, post-debond care. Discomfort subtheme: Satisfaction with treatment not affected by this as they expected it as part of the treatment journey. Function and oral hygiene: Difficulty with eating certain foods reported, but most participants found ways around this by the end of treatment. Aesthetics: Adult orthodontics being widely accepted now made this easier to deal with. Some participants from a private setting said they would not have undergone treatment had metal fixed appliances been the only treatment option. Post-debond care: Participants found having retention reviews reassuring, which improved their overall treatment satisfaction.

## Data Availability

No new data were created or analyzed in this study.

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
