# Peer review of "Oral Health-Related Quality of Life (OHRQoL), Pain and Side Effects in Adults Undergoing Different Orthodontic Treatment Modalities: A Systematic Review and Meta-Analysis"

_healthcare, 2025, doi:10.3390/healthcare13243317_

Round 1

Reviewer 1 Report

Comments and Suggestions for Authors

Dear respected authors. By addressing the following points, the clarity and impact of your manuscript can be significantly improved, facilitating better understanding and engagement from readers and reviewers.

  1. The current title and background are not fully aligned, and the objective of the review is not clearly conveyed. The title suggests an evaluation of OHRQoL, pain, and side effects among adults undergoing different orthodontic treatment modalities. However, the background states that the study aims to identify differences in patient experiences before and after treatment using removable aligners or fixed labial/lingual appliances. This phrasing is unclear and may imply both a longitudinal comparison (pre- vs post-treatment) and a comparison between different appliance types.
  2. You state that “the findings may not extrapolate to all clinical settings; participants were adults treated mainly in teaching hospitals with mild malocclusions.” If the included population is predominantly adults with mild malocclusions, the title should more accurately reflect this limitation. As currently written, the title implies that the conclusions apply broadly to all adults undergoing different orthodontic treatment modalities, regardless of case complexity.

I recommend modifying the title or, at minimum, adding a clarifying phrase to ensure it accurately represents the study population. This will help prevent overgeneralization and improve the precision and transparency of the manuscript.

Author Response

Dear Reviewer 1,

Comments: 1. The current title and background are not fully aligned, and the objective of the review is not clearly conveyed. The title suggests an evaluation of OHRQoL, pain, and side effects among adults undergoing different orthodontic treatment modalities. However, the background states that the study aims to identify differences in patient experiences before and after treatment using removable aligners or fixed labial/lingual appliances. This phrasing is unclear and may imply both a longitudinal comparison (pre- vs post-treatment) and a comparison between different appliance types.

Response: We acknowledge this comment and have as a result modified the aims to improve clarity and to better align with the title, to the following:

Thus, the present study aimed to identify the differences between experiences, in terms of oral health related quality of life, pain, side-effects and/or other complications, of adults undergoing orthodontic treatment using removable aligners and fixed labial or lingual appliances.

Comments: 2. You state that “the findings may not extrapolate to all clinical settings; participants were adults treated mainly in teaching hospitals with mild malocclusions.” If the included population is predominantly adults with mild malocclusions, the title should more accurately reflect this limitation. As currently written, the title implies that the conclusions apply broadly to all adults undergoing different orthodontic treatment modalities, regardless of case complexity.

Response: We acknowledge this comment and have as a result modified the limitations section to improve clarity, to the following: The findings may not be extrapolated to all clinical settings in which these appliances are used, for example adolescent patients. The present study only evaluated the evidence in relation to adults seeking orthodontic treatment, to identify any differences between their experiences with these appliances.

Comment: I recommend modifying the title or, at minimum, adding a clarifying phrase to ensure it accurately represents the study population. This will help prevent overgeneralization and improve the precision and transparency of the manuscript.

Response: We acknowledge this comment and have as a result modified in regards to the above comment 1.

Reviewer 2 Report

Comments and Suggestions for Authors

The manuscript entitled “Oral Health Related Quality of Life (OHRQoL), Pain and Side Effects in Adults Undergoing Different Orthodontic Treatment Modalities: A Systematic Review and Meta-Analysis” is overall well-organized; however, several revisions are required to enhance its clarity, methodological rigor, and interpretative value.

  1. Abstract
  • The research objective should be stated more clearly, with explicit specification of the outcome measures being compared.
  • The study design should be clearly identified as a systematic review and meta-analysis, and the databases searched as well as the search period should be explicitly reported.
  • Please include a brief statement addressing heterogeneity and its influence on the results.

  1. Methods Section (Line 84)

In the sentence beginning with “Study design: The following were considered eligible: Randomized controlled trials (RCTs)…,” the heading “Study design:” should be removed to improve the flow and clarity of the text.

  1. Results Section
  • In Figure 1, please provide clearer descriptions of the reasons for exclusion during the screening process. Although Appendix 2 presents the detailed reasons, a concise summary should also be integrated into the figure to enhance transparency.
  • It is recommended to include information on the countries in which the included studies were conducted.
  1. Discussion
  • The discussion should elaborate on the clinical implications of the findings.
  • Additionally, please provide a more detailed section on future research directions to strengthen the scholarly contribution of the manuscript.
  1. Author Contributions

Please expand the description of each author’s specific contributions to the work in accordance with authorship guidelines.

Author Response

Dear Reviewer 2,

Comments: The manuscript entitled “Oral Health Related Quality of Life (OHRQoL), Pain and Side Effects in Adults Undergoing Different Orthodontic Treatment Modalities: A Systematic Review and Meta-Analysis” is overall well-organized; however, several revisions are required to enhance its clarity, methodological rigor, and interpretative value.

  1. Abstract
  • Comments: The research objective should be stated more clearly, with explicit specification of the outcome measures being compared.

Response: We acknowledge this comment and have as a result modified as per reviewer 1 comment too, to the following: Thus, the present study aimed to identify the differences between experiences, in terms of oral health related quality of life, pain, side-effects and/or other complications, of adults undergoing orthodontic treatment using removable aligners and fixed labial or lingual appliances.

  • Comments: The study design should be clearly identified as a systematic review and meta-analysis, and the databases searched as well as the search period should be explicitly reported.

Response: Thank you for this comment but we feel both these suggestions are already included under the headings of ‘Protocol and Registration’ where we state: The current systematic review and meta-analysis was carried out according to the Preferred Reporting Items for Systematic Reviews and Meta-Analyses (PRISMA) guidelines [8].

In terms of the search period, this is included under the section ‘Information sources, search strategy and selection criteria’ as follows: Databases searched from inception until 3rd May 2023.

  • Comments: Please include a brief statement addressing heterogeneity and its influence on the results.

Response: We thank the reviewer for this comment and have correspondingly added such brief statements in relation to the results, as follows:

The two aligner studies were homogenous (I2 = 0%) adding significantly to the validity of the findings.

The standardised mean difference in pain VAS scores between 24 hours and seven days was estimated -6.4 (95% CI: -10.42, -2.36) for labial appliances. The test for significant differences between them was not confirmed (χ2=2.89, p=0.09). The heterogeneity for this comparison was moderate (I2=65.4%), suggesting a greater degree of caution in interpreting the findings from the studies evaluating labial appliances.

The standardised mean difference for dental self-confidence tested across two publications indicated a significant increase between pre- and post-treatment scores of 2.18 (95% CI:0.87, 3.5) (z=3.25, p=0.001) (Figure 7). However, the heterogeneity was high between the studies (I2=97%), suggesting a greater degree of caution in interpreting the findings from these studies.

A reduction in aesthetic concern was not significant when the data were pooled (Figure 8), with scores of -1.71 (95% CI: -3.46, 0.04) (z=1.91, p=0.06). Heterogeneity across the studies was very high (I2=99%), again suggesting a greater degree of caution in interpreting the findings from these studies.

  1. Methods Section (Line 84)

Comments: In the sentence beginning with “Study design: The following were considered eligible: Randomized controlled trials (RCTs)…,” the heading “Study design:” should be removed to improve the flow and clarity of the text.

Response: We acknowledge this comment and have removed the term ‘study design’ from the section referring to the PICO selection criteria:

  1. Results Section
  • Comments: In Figure 1, please provide clearer descriptions of the reasons for exclusion during the screening process. Although Appendix 2 presents the detailed reasons, a concise summary should also be integrated into the figure to enhance transparency.

Response: Thank you for raising this comment and have added the following as the principle reason for their exclusion after screening: Failed to meet the PICO selection criteria and as the reviewer acknowledges the precise detail in provided in Appendix 2

  • Comments: It is recommended to include information on the countries in which the included studies were conducted.

Response: Whilst we acknowledge this comment, we do not feel it is necessary as the included studies were not selected on this basis but rather on their need to meet the identified PICO selection criteria and the included studies do not all universally provide this information.

  1. Discussion
  • Comments: The discussion should elaborate on the clinical implications of the findings.

Response: We thank the reviewer for his comment but do not feel there is any further need to expand on the already extensive discussion of the results and their implications but feel that the acknowledgement of the comment below serves to highlight the need for further research in the field.

  • Comments: Additionally, please provide a more detailed section on future research directions to strengthen the scholarly contribution of the manuscript.

Response: We acknowledge this comment and have as a result have added the following sentence to the discussion: Future research projects should aim to utilise well defined populations, with valid and commonly reported outcome measures at agreed timepoints to limit study heterogeneity and increase the evidence-base in this expanding adult clinical practice.

  1. Author Contributions

Comments: Please expand the description of each author’s specific contributions to the work in accordance with authorship guidelines.

Response: We acknowledge this comment but it was not an apparent requirement of the submission. However for completeness we include it here as follows:

Authors' contributions: AJ: conceptualisation, methodology, data curation, review and editing of manuscript. BD, HB and CS: investigation, data curation, writing original draft, All authors have read and approved the final manuscript.

Reviewer 3 Report

Comments and Suggestions for Authors

Dear authors, your manuscript is very interesting.
Here are some suggestions to improve certain parts and make it worthy of publication
In the introduction, you need to add more information about the quality of life of patients with orthodontic devices and you need to add the limitations and side effects that these devices can have on people's lives.
The materials and methods section is well described, as is the results section.
In the discussion, more articles should be added that are already present in the literature and that have addressed a similar topic.
The conclusions should be expanded.
The bibliography is good.

Author Response

Dear Reviewer 3,

Reviewer 3

Comments:

Dear authors, your manuscript is very interesting.
Here are some suggestions to improve certain parts and make it worthy of publication

In the introduction, you need to add more information about the quality of life of patients with orthodontic devices and you need to add the limitations and side effects that these devices can have on people's lives.

Response: We acknowledge this comment and have as a result modified the Introduction to include the following: An in-depth understanding of the adult patient experiences with the different range of appliances now available would allow clinicians to provide more relevant information to patients and in turn, facilitate the development of a patient-centred approach to providing more optimal care and informed consent. However, to-date there is relatively very little understanding of the impact of these appliance choices on their daily lives, with only two quantitative studies comparing the three appliance systems in relation to pain and oral health related quality of life (OHRQoL) in adults, with their findings being inconclusive [8.9]. 

In contrast, a greater number of studies exist to compare clear aligner therapy with fixed labial appliances. They report participants treated with the former report less pain, eating difficulties and an overall lower in-treatment negative experience, during the early stages of treatment [10-13]. In contrast, comparison of fixed labial and lingual appliances does not appear to demonstrate any significant differences in pain experience but the latter were associated with greater oral discomfort, dietary change, swallowing and speech disturbances, and social problems than those in labial appliance group [14,15].

Comments: The materials and methods section is well described, as is the results section.

In the discussion, more articles should be added that are already present in the literature and that have addressed a similar topic.

Response: We acknowledge this comment and feel with the requested expansion of the Introduction [above], the existing near 4-pages of discussion covers the findings and any shortcomings in sufficient detail.

Comments: The conclusions should be expanded

Response: We acknowledge this comment but not feel there is any further need to expand the existing 5 bullet points made in the conclusion.

Comments: The bibliography is good.

Round 2

Reviewer 3 Report

Comments and Suggestions for Authors

The authors have provided satisfactory explanations to my questions and I do not request any further changes.
Congratulations.